# Effects of DNA Topology on Transcription from rRNA Promoters in *Bacillus subtilis*

**DOI:** 10.3390/microorganisms9010087

**Published:** 2021-01-01

**Authors:** Petra Sudzinová, Milada Kambová, Olga Ramaniuk, Martin Benda, Hana Šanderová, Libor Krásný

**Affiliations:** Institute of Microbiology of the Czech Academy of Sciences, 142 00 Prague, Czech Republic; petra.sudzinova@biomed.cas.cz (P.S.); mkambova@gmail.com (M.K.); ramaniuk@biomed.cas.cz (O.R.); mb.martin.benda@gmail.com (M.B.); sanderova@biomed.cas.cz (H.Š.)

**Keywords:** *Bacillus subtilis*, transcription, ribosomal RNA, DNA topology

## Abstract

The expression of rRNA is one of the most energetically demanding cellular processes and, as such, it must be stringently controlled. Here, we report that DNA topology, i.e., the level of DNA supercoiling, plays a role in the regulation of *Bacillus subtilis* σ^A^-dependent rRNA promoters in a growth phase-dependent manner. The more negative DNA supercoiling in exponential phase stimulates transcription from rRNA promoters, and DNA relaxation in stationary phase contributes to cessation of their activity. Novobiocin treatment of *B. subtilis* cells relaxes DNA and decreases rRNA promoter activity despite an increase in the GTP level, a known positive regulator of *B. subtilis* rRNA promoters. Comparative analyses of steps during transcription initiation then reveal differences between rRNA promoters and a control promoter, P*veg*, whose activity is less affected by changes in supercoiling. Additional data then show that DNA relaxation decreases transcription also from promoters dependent on alternative sigma factors σ^B^, σ^D^, σ^E^, σ^F^, and σ^H^ with the exception of σ^N^ where the trend is the opposite. To summarize, this study identifies DNA topology as a factor important (i) for the expression of rRNA in *B. subtilis* in response to nutrient availability in the environment, and (ii) for transcription activities of *B. subtilis* RNAP holoenzymes containing alternative sigma factors.

## 1. Introduction

Bacterial cells need to adapt to environmental changes. In nutrient-rich environments, cells grow and divide rapidly, and this requires a large number of ribosomes to satisfy the need for new proteins. In nutritionally poor environments, the synthesis of new ribosomes stops. As the production of new ribosomes is energetically costly for the cell, it must be tightly regulated. The number of ribosomes in the cell is regulated mainly on the level of transcription initiation of ribosomal RNA (rRNA) [1].

Transcription initiation can be divided into several steps. First, when the RNA polymerase (RNAP) holoenzyme (the core RNAP subunits [α2ββ´ω] in complex with a σ factor) binds to specific DNA sequences, promoters, it forms the closed complex where DNA is still in the double-helical form [2]. The specificity of RNAP for promoter sequences is provided by the σ factor [3,4,5,6]. Subsequently, this complex isomerizes and forms the open complex where the two DNA strands are unwound, and the transcription bubble is formed. At this stage, initiating nucleoside triphosphates NTPs (iNTPs) can enter the active site and transcription can begin. RNAP then leaves the promoter and enters the elongation phase of transcription [7].

In bacteria, the concentrations of iNTPs act as key regulators of transcription and directly affect RNAP at some promoters. These promoters form relatively unstable open complexes where the time window available to iNTPs to penetrate into the active site and initiate transcription is relatively short. The higher the concentration of the respective iNTP, the higher the chance that it penetrates into the active site while the transcription bubble is still open. Hence, increases in intracellular concentrations of iNTPs stimulate transcription whereas low levels of iNTPs result in inefficient transcription initiation [8,9,10].

rRNA promoters are prime examples of where transcription initiation is regulated by the concentration of the iNTP. In *Bacillus subtilis*, a model soil-dwelling, spore-forming Gram-positive bacterium, the iNTP of the tandem rRNA promoters of all 10 rRNA operons is exclusively GTP [11]. The GTP level in *B. subtilis* is affected by (p)ppGpp, an alarmone that is produced at times of stress, such as amino acid starvation or heat shock. (p)ppGpp inhibits GuaB, the first enzyme in the de novo GTP biosynthesis pathway, which results in decreased GTP levels and increased ATP levels as more of the last common intermediate for the synthesis of both GTP and ATP, inosine monophosphate (IMP), is now available for ATP synthesis only [12,13]. By affecting the GTP level (p)ppGpp indirectly affects the activity of rRNA promoters in *B. subtilis* [14,15,16]. This might be similar in other Gram-positive microorganisms, such as *Staphylococcus aureus*, where the GTP concentration ([GTP]) affects rRNA promoter activity under stringent conditions [17].

Another important factor for transcription initiation in bacteria is the topological state of DNA, i.e., the levels of supercoiling. DNA in the cells is usually underwound and this results in negative supercoiling [18]. Negative supercoiling then helps RNAP to melt DNA in promoter regions. In general, bacterial cells display more pronounced negative supercoiling in exponential than in stationary phase of growth and initiation from a number of promoters is sensitive to this parameter [19,20,21,22,23].

Here, we investigated how the activity of rRNA promoters in *B. subtilis* changes when the cells transition from exponential to stationary phase. These promoters depend on the primary σ factor, σ^A^. We show that their activity decreases during the transition and this correlates with a decrease in the GTP concentration. Nevertheless, there is a point in the process where the level of GTP does not decrease any further but the activity of rRNA promoters does. We show that besides [GTP], *B. subtilis* rRNA promoters are regulated by the level of their supercoiling, and we dissect the effects of supercoiling on the formation of closed and open complexes, thereby providing mechanistic insights into the process. Finally, we show that supercoiled (SC) DNA is a more efficient template for transcription for all alternative σ factors tested with the exception of σ^N^, a recently discovered sigma factor encoded on the pBS32 plasmid of the NCIB 3610 strain [24,25]. In summary, a newly updated model of *B. subtilis* promoter regulation is presented here.

## 2. Materials and Methods 

### 2.1. Media and Growth Conditions

Cells were grown at 37 °C, either in LB or in rich MOPS supplemented with 20 amino acids: 50 mM MOPS (pH 7.0), 1 mM (NH_4_)_2_SO_4_, 0.5 mM KH_2_PO_4_, 2 mM MgCl_2_, 2 mM CaCl_2_, 50 μM MnCl_2_, 5 μM FeCl_3_, amino acids (50 μg/mL each), and 0.4% glucose. Antibiotics used: ampicillin 100 μg/mL, chloramphenicol 5 μg/mL, novobiocin 5 μg/mL, and rifampicin 2 μg/mL. Strains used are listed in Table 1.

### 2.2. Bacterial Strains

### 2.3. Determination of ATP, GTP, and ppGppconcentrations

Strains of *B. subtilis* (LK134, for *rrn*B P1 and LK135 for P*veg*) were grown in the MOPS 20 AA medium supplemented with [^32^P] KH_2_PO_4_ (100 μCi/mL) until early exponential phase (OD_600_ ~0.3). Samples were taken until 250 min after OD_600_ ~0.3 (time 0). Samples (100 μL) were pipetted into 100 μL 11.5 M formic acid, vortexed, left on ice for 20 min, and stored overnight at −80 °C [29]. After microcentrifugation (5 min, 4 °C) to remove cell debris, the samples (5 μL) were spotted on TLC (thin-layer chromatography) plates (Polygram^®^CEL 300 PEI, purchased from Macherey-Nagel), developed in 0.85 M (for ATP and GTP) or 1.5 M (for ppGpp) KH_2_PO_4_ (pH 3.4) and quantified by phosphorimaging. The identities of ATP, GTP, and ppGpp were verified by comparison with commercial preparations of these compounds run in parallel and visualized by UV shadowing [8]. 

To determine the relative ATP/GTP concentrations after novobiocin treatment, LK134 was grown to OD_600_ ~0.3 (time 0) in medium supplemented with [^32^P] H_3_PO_4_ (100 μCi/mL), and at time 5 min treated with novobiocin (5 μg/mL). Samples were taken at time points 0, 5, 10, 20, and 30 min and processed in the same way as above.

### 2.4. Promoter Activity Monitored by Quantitative Primer Extension (qPE)

Promoter constructs were fused to *lacZ* and activities were assayed by primer extension of the short-lived *lacZ* mRNA that allows to observe rapid decreases in promoter activity in time. The experiments were conducted as described in [15]. Typically, 1 mL of cells was pipetted directly into 2 mL phenol/chloroform (1:1) and 0.25 mL lysis buffer (50 mM Tris–HCl pH 8.0, 500 mM LiCl, 50 mM EDTA pH 8.0, 5% SDS). After brief vortexing, the recovery marker (RM) was added. The RM RNA was made from *B. subtilis* strain LK41 as described in [15]. This was followed by immediate sonication. Water was then added to increase the aqueous volume to 6 mL to prevent precipitation of salts, followed by two extractions with phenol/chloroform, two precipitations with ethanol, and suspension of the pellet in 20–50 μL 10 mM Tris–HCl, pH 8.0.

Primer extension was performed with M-MLV reverse transcriptase as recommended by the manufacturer (Promega) with 1–10 μL purified RNA. The ^32^P 5’-labeled primer (#2973) hybridized 89 nt downstream from the junction of the promoter fragment used for the creation of the *lacZ* fusion. Samples were electrophoresed on 7 M urea 5.5% or 9% polyacrylamide gels. The gels were exposed to Fuji Imaging Screens. The screens were scanned with Molecular Imager FX (Bio-Rad, Berkeley, CA, USA) and were visualized and analysed using the Quantity One software (Bio-Rad), and normalized to cell number (OD_600_) and RM.

### 2.5. Promoter Activity Monitored by RT–qPCR

*rrn*B P1 and P*veg* promoters were fused to the marker *lacZ* gene (LK134 and LK135), yielding identical transcripts. The strains were grown to exponential phase (OD_600_ ~0.5)—time point 0. Each culture was then divided into two flasks. Cells in one flask were treated with novobiocin (5 μg/mL) and cells in the other flask were left non-treated. At time points 0, 10, 20, and 30 min, 2 mL of cells were withdrawn and treated with RNAprotect Bacteria reagent (QIAGEN, Hilden, Germany), pelleted and immediately frozen. RNA was isolated with RNeasy Mini Kit (QIAGEN) and recovery marker RNA (RM RNA) was added at the time of extraction to control for differences in degradation and pipetting errors during extraction. The RM RNA was prepared from *B. subtilis* strain LK41 as for qPE. Finally, RNA was DNase treated according to manufacturers’ instructions (TURBO DNA-free Kit, Ambion). Total RNA was then reverse transcribed to cDNA with reverse transcriptase (SuperScript™ III Reverse Transcriptase, Invitrogen, Waltham, MA, USA) using primer #2973 that targets *lacZ* (both in the test mRNA and RM). This was followed by qPCR in a LightCycler 480 System (Roche Applied Science, Penzberg, Germany) containing LightCycler^®^ 480 SYBR Green I Master and 0.5 μM primers (each). RM cDNA was amplified with primers #2974 and #2973, and the test lacZ cDNA with primers #2975 and #2973. Sequences of primers were originally published in [15]. The final data were normalized to RM and the amount of cells (OD_600_).

### 2.6. ^3^H-Incorporation in Total RNA

This experiment was conducted as described previously [30]. Briefly, strain LK134 was grown in LB medium to OD_600_ ~0.3 (early exponential phase). Newly synthesized RNA in the cells was labeled with ^3^H-uridine (1 μCi/mL) (cold [non-radioactive] uridine was added to a final concentration of 100 μM); time point 0. The bacterial culture was divided into three flasks—non-treated, treated with novobiocin (5 μg/mL), and treated with rifampicin (2 μg/mL), respectively (time point 5). At 0, 5, 10, 20 and 30 min, 100 μL and 250 μL of cells were withdrawn to measure cell density and determine ^3^H-incorporation, respectively. The 250 μL cell sample was mixed with 1 mL of 10% trichloroacetic acid (TCA) and kept on ice for at least 1 h. Thereafter, each sample was vacuum filtered, using Glass Microfiber Filters (Whatman, Little Chalfont, UK), washed twice with 1 ml of 10% TCA and three times with 1 mL of ethanol. The filters were dried, scintillation liquid was added, and the radioactivity was measured. The signal was normalized to cell density (OD_600_).

### 2.7. RNAP Levels in Time

Cells (strain LK134) were grown in LB rich medium to OD_600_ 0.3 (time point 0). Subsequently, every 30 min 10 mL of cells were pelleted and OD_600_ was measured. Pellets were washed with Lysis Buffer (20 mM Tris-HCl, pH 8, 150 mM KCl, 1 mM MgCl_2_) and frozen. Next day, pellets were resuspended in Lysis Buffer (100–500 μL, according to the size of pellet) and disrupted by sonication 2 × 1 min, with 1 min pause on ice between the pulses. After centrifugation (5 min, 4 °C) to remove cell debris, the amounts of proteins were measured with the Bradford protein assay and 5 μg was resolved by SDS-PAGE and analyzed by Western blotting, using mouse monoclonal antibodies against the β subunit of RNAP (clone name 8RB13, dilution 1:1000, Genetex, Irvine, CA, USA) and anti-mouse secondary antibody conjugated with HRP (dilution 1:800,000, Sigma, Munich, Germany). Subsequently, the blot was incubated for 5 min with SuperSignal^TM^ West Femto PLUS Chemiluminiscent substrate (Thermo scientific, Waltham, MA, USA), exposed on film and developed.

### 2.8. Proteins and DNA for Transcription In Vitro

#### 2.8.1. Strain Construction

Genes encoding σ^B^, σ^E^, σ^F^ and σ^H^ were amplified from genomic wt DNA by PCR with Expand High Fidelity PCR System (Roche) with respective primers (Table 1, Material and Methods section) and cloned into pET-22b(+) via *Nde*I*/Xho*I restriction sites and verified by sequencing. Primers for cloning of σ^E^ were designed for the active form of protein, as its first 27 AA are in the cell posttranslationally removed [31,32]. The resulting plasmids were transformed into expression strain BL21(DE3), yielding strains LK1207 (σ^B^), LK2580 (σ^E^), LK1425 (σ^F^), and LK1208 (σ^H^). His-SUMO-σ^N^ fusion protein in an expression plasmid pBM05 [25] was transformed to BL21(DE3), resulting in strain LK2531.

#### 2.8.2. Protein Purification

Wild type RNAP, containing a His10x-tagged β’ subunit was purified from LK1723 as described [26]. 

The SigA subunit of RNAP (LK22) was overproduced a purified as described [27].

σ^B^, σ^E^, σ^F^, σ^H^ expression strains were grown to OD_600_ ~0.5 when IPTG was added to a final concentration of 0.8 mM. Cells were allowed to grow for 3 h at room temperature, cells were harvested, washed and resuspended in P buffer (300 mM NaCl, 50 mM Na_2_HPO_4_, 3 mM β-mercaptoethanol, 5% glycerol). Cells were then disrupted by sonication and the supernatant was mixed with 1 mL Ni-NTA agarose (QIAGEN, Hilden, Germany) and incubated for 1 h at 4 °C with gentle shaking. Ni-NTA agarose with the bound protein was loaded on a Poly-Prep^®^ Chromatography Column (Bio-Rad, Berkeley, CA, USA), washed with P buffer and subsequently with the P buffer with the 30 mM imidazole. The protein was eluted with P buffer containing 400 mM imidazole and fractions containing σ factor were pooled together and dialyzed against storage buffer (50 mM Tris-HCl, pH 8.0, 100 mM NaCl, 50% glycerol and 3 mM β-ME). The proteins were stored at −20 °C.

σ^D^ was purified from inclusion bodies as described in [28].

Cells containing the plasmid for overproduction of σ^N^ were grown to OD_600_ ~0.5 and IPTG was added to final concentration 0.3 mM. Cells were then allowed to grow for 3 h at room temperature; afterwards the cells were harvested, washed, and resuspended in P buffer. All purification steps were done in P2 buffer (the same composition as P buffer, but pH 9.5). Cells were then disrupted by sonication and the supernatant was mixed with 1 mL Ni-NTA agarose (QIAGEN) and incubated for 1 h at 4 °C with gentle shaking. Ni-NTA agarose with the bound His-SUMO-σ^N^ was loaded on a Poly-Prep^®^ Chromatography Column (Bio-Rad), washed with P2 buffer and subsequently with the P2 buffer with the 30 mM imidazole. The protein was eluted with P2 buffer containing 400 mM imidazole and fractions containing His-SUMO-σ^N^ were pooled together and dialyzed against P2 buffer.

The SUMO tag was subsequently removed by using SUMO protease (Invitrogen). The cleavage reaction mixture was again mixed with the 1 mL Ni-NTA agarose and allowed to bind for 1 h at 4 °C and centrifuged to pellet the resin. Supernatant was removed, the resin was washed once more with P2 buffer with 3 mM β-ME. The supernatants (containing σ^N^) were pooled together and dialysed against storage P2 buffer (P2 buffer and 50% glycerol). The protein was stored at −20 °C.

The purity of all proteins was checked by SDS-PAGE.

#### 2.8.3. Promoter DNA Construction

Promoter regions of alternative σ-dependent genes were amplified from genomic wt DNA of *B. subtilis* with primers listed in Table 2 (Material and Methods section) by PCR. All fragments were then cloned into p770 (pRLG770 [33]) using *EcoR*I/*Hind*III restriction sites and transformed into DH5α. All constructs were verified by sequencing.

Supercoiled plasmids (SC) were obtained using the Wizard^®^ Plus Midipreps DNA Purification System, for higher yields Wizard^®^ Plus Maxipreps DNA Purification System (both Promega, Madison, WI, USA) were used and subsequently phenol-chloroform extracted, precipitated with ethanol, and dissolved in water. Aliquots of plasmids were linearized with the *Pst*I restriction enzyme (TaKaRa, Saint-Germain-en-Laye, Francie), resulting in linear form (LIN), and again precipitated with ethanol to remove salts.

The state of DNA topology (linear, supercoiled) was checked on agarose gels.

#### 2.8.4. List of Primers

### 2.9. Transcription In Vitro

Transcription experiments were performed with the *B. subtilis* RNAP core reconstituted with a saturating concentration of σ^A^ (ratio 1:5) in storage buffer (50 mM Tris-HCl, pH 8.0, 0.1 M NaCl, 50% glycerol) for 15 min at 30 °C. The 1:5 ratio was used also for σ^B^, σ^D^, σ^E^, σ^F^, and σ^H^. For σ^N^, the ratio was 1:8. Multiple round transcription reactions were carried out in 10 μL reaction volumes with 30 nM RNAP holoenzyme. The transcription buffer contained 40 mM Tris-HCl pH 8.0, 10 mM MgCl_2_, 1 mM dithiothreitol (DTT), 0.1 mg/mL BSA and 150 mM KCl, and all four NTPs and 2 μM radiolabeled [α-^32^P] UTP.

In K_GTP_ determination experiments, the amount of DNA (SC or LIN form) was 100 ng, ATP, CTP were 200 μM; UTP was 10 μM and GTP was titrated from 0 to 2000 μM. To determine the affinity of RNAP to DNA, ATP, CTP were at 200 μM; UTP was 10 μM, GTP was 1000 μM and DNA (SC/LIN) was titrated from 0 to 900 ng per reaction. In reactions with alternative σ, DNA (SC or LIN form) was 100 ng, CTP were at 200 μM; UTP was 10 μM and GTP/ATP was 1000 μM, depending on the identity of the base in the +1 position of the transcript. 

All transcription reactions were allowed to proceed for 15 min at 30 °C and then stopped with equal volumes of formamide stop solution (95% formamide, 20 mM EDTA, pH 8.0). Samples were loaded onto 7 M urea-7% polyacrylamide gels and electrophoresed. The dried gels were scanned with Molecular Imager FX (Bio-Rad) and were visualized and analysed using the Quantity One software (Bio-Rad).

## 3. Results

### 3.1. The Activity of rrnB P1 Decreases during Entry into Stationary Phase

As the main model rRNA promoter, we selected the *rrn*B P1 promoter as it is one of the best-characterized rRNA promoters in *B. subtilis* that is regulated by [iNTP], [11,34,35,36]. Furthermore, the dynamic range of the activity of *rrn*B P1 is wide, which facilitated the design and interpretation of the experiments. As the main control promoter, we selected the strong P*veg* promoter that forms stable open complexes and is saturated with a relatively low level of its iNTP. This promoter drives transcription of the *veg* gene that is involved in biofilm formation [37,38]. Promoter sequences are shown in Figure 1A.

To monitor promoter activities, we used core promoter-*lacZ* fusions. The endogenous copy of P*veg* initiates transcription with ATP (+1A). Here, we used a +1G variant of P*veg* so that both transcripts (from *rrn*B P1-*lacZ* and P*veg*-*lacZ*) were identical, excluding any effects due to, e.g., potentially differential decay of the transcripts. The +1G P*veg* promoter variant behaves identically with the +1A variant [11]. Throughout the study, promoter activity was determined by quantitative primer extension (qPE) or reverse transcription followed by quantitative PCR (RT-qPCR).

We used defined rich MOPS medium to grow the cells and measured (i) relative GTP level ([GTP]) and (ii) relative promoter activity (*rrn*B P1 and P*veg*) from early exponential phase till approximately two hours into stationary phase by qPE (Figure 1). 

We detected a moderate decrease in [GTP] already during exponential phase (Figure 1B). This moderate decrease was followed by a precipitous decline during the transition between the two phases. This correlated with a sharp spike in the (p)ppGpp level (Appendix A). However, early on in the stationary phase, [GTP] even slightly increased and then remained at the same level till the end of the experiment. The activities of both *rrn*B P1 and P*veg* decreased during the time course of the experiment—the activity of the former more than of the latter, consistent with the behavior of these promoters as reported in previous studies [10,11]. 

Surprisingly and interestingly, the activity *rrn*B P1 decreased even after the relative GTP concentration had been stabilized at a constant level. This strongly suggested that another mechanism, besides rRNA promoter regulation by [GTP], exists in the cell. DNA supercoiling is known to change between growth phases, typically the negative supercoiling from exponential phase becomes more relaxed in stationary phase, as demonstrated for *Escherichia coli* [39] and also *B. subtilis* [40]. Also, we noticed that the activity of P*veg* significantly decreased, although the decrease was not as pronounced as that of the ribosomal promoter. As DNA topology is an important factor for gene expression regulation, we decided to address the potential of *B. subtilis* rRNA promoters to be regulated by the level of supercoiling.

### 3.2. Chromosome Relaxation Inhibits Total RNA Synthesis In Vivo

To test whether DNA topology could affect rRNA expression in vivo, we used novobiocin. Novobiocin is an antimicrobial compound that binds to the β subunit of gyrase and blocks its function by inhibiting ATP hydrolysis [41,42,43]. Gyrase relieves tension in DNA caused by transcribing RNAPs or helicases by creating more negatively supercoiled DNA. Hence, the inhibition of gyrase causes DNA in the cell to be more relaxed [44]. 

In this experiment, we first used total RNA as a proxy for rRNA synthesis as in exponential phase most of RNA synthesis comes for rRNA operons (~80% of RNA in cell is rRNA and tRNA [29,45]). We treated early-exponentially growing cells (OD_600_ ~0.3) with novobiocin or mock-treated them, and measured the rates of total RNA synthesis by following incorporation of radiolabeled ^3^H-uridine into RNA. As a positive control, where we expected cessation of RNA synthesis, we treated cells with rifampicin, a well-characterized inhibitor of bacterial RNAP.

Figure 2 shows that in the presence of novobiocin the synthesis of total RNA decreased/stopped, similarly as in the presence of rifampicin, suggesting that relaxation of the chromosome affects total RNA synthesis in the cell (Figure 2A).

### 3.3. Novobiocin-Induced Relaxation of DNA Affects the Activity of rrnB P1 In Vivo

Next, by RT-qPCR we monitored the response of *rrn*B P1 and P*veg* to novobiocin treatment, using the same conditions as in the previous experiment. We grew cells carrying the appropriate fusions (*rrn*B P1-*lacZ* (LK134) and P*veg*-*lacZ* (LK135)) to early-exponential phase (OD_600_ ~0.3) and either treated them with novobiocin or mock-treated them. In the case of *rrn*B P1, the promoter activity decreased after novobiocin treatment (as opposed to mock treatment), but in the case of P*veg*, the promoter activity displayed the same moderate decline regardless of the novobiocin treatment, suggesting that *rrn*B P1 is more sensitive to changes in DNA topology (Figure 2B,C).

We also measured the GTP levels in novobiocin treated cells. We observed that novobiocin-induced relaxation resulted in a massive increase in the GTP level in cell (Figure 2D). The levels of ATP increased only slightly (Appendix A). Thus, the activity of *rrn*B P1 and the level of GTP became uncoupled. These experiments suggested that DNA topology might affect the activity rRNA promoters, but it was also possible that unknown, secondary effects of the novobiocin treatment could be the cause.

### 3.4. Changes in DNA Topology Affect the Affinity of RNAP for iNTP In Vitro

To test directly whether DNA topology affects the activity of rRNA promoters, we performed in vitro transcription experiments. We had speculated that the in vivo decrease in the activity of *rrn*B P1 during stationary phase and in response to novobiocin treatment could be due to altered affinity of RNAP for iGTP at this promoter (induced by changes in supercoiling levels): the GTP level does not change but the open promoter becomes less stable, requiring more iGTP for maximal transcription. To address this hypothesis experimentally, we performed in vitro transcriptions with defined components. We used promoter core variants of *rrn*B P1 and P*veg* cloned in the pRLG770 plasmid [11] (for details see Table 1 in Material and Methods section). The DNA templates were used in two different topological forms—in the negatively supercoiled plasmid form (SC), and in the relaxed form (LIN), using the same DNA construct but linearized with the *Pst*I restriction enzyme (Appendix A). 

We performed multiple round transcriptions in vitro with increasing [GTP] (Figure 3). The GTP concentration required for half-maximal transcription (K_GTP_) was used as a measure of the affinity of RNAP for iGTP at the promoter. A characteristic of rRNA promoters is their requirement for relatively high levels of iGTP for maximal transcription (due to unstable open complexes), reflected in high values of K_GTP_ in vitro. P*veg*, to the contrary, has a low value of K_GTP_.

Experiments with SC templates confirmed previously published results [46], the K_GTP_ for *rrn*B P1 was 277 ± 24 μM, and for P*veg* 36 ± 9 μM. Experiments with the LIN templates then revealed that K_GTP_ values for both promoters increased (*rrn*B P1 = 440 ± 25 μM, P*veg* = 511 ± 78 μM). In the case of *rrn*B P1 the K_GTP_ increased from SC to LIN ~1.5x, and in the case of P*veg* K_GTP_ ~14x. Surprisingly, the K_GTP_ value of LIN P*veg* was even higher than the value for *rrn*B P1 (Figure 3B). 

Importantly, the experiments showed that the strength (the maximal level of transcription) of the *rrn*B P1 promoter dramatically decreased on the LIN template whereas in the case of P*veg* the maximal level of transcription was comparable for both types of the template (Figure 3A, primary data), confirming the hypothesis that DNA relaxation decreases the activity of *rrn*B P1 more than the activity of P*veg*.

As the preceding experiments were done with the core version of the *rrn*B P1 promoter, we also decided to use an extended version of the promoter region to assess whether the surrounding sequence has significant effects. Therefore, we used a DNA fragment containing both *rrn*B P1 and *rrn*B P2 promoters in their native tandem arrangement. Each of them contained their respective native -60 to -40 regions encompassing the UP elements. UP elements are A/T-rich sequences that enhance promoter activity by binding the C-terminal domains of α-subunits of RNAP [47,48,49]. Although their stimulatory effect on rRNA promoters in *B. subtilis* is less pronounced than, e.g., in *E. coli* (~30x), it is still significant [11]. Experiments with these promoter versions yielded virtually the same results as with the core version (Figure 3C). The K_GTP_ for *rrn*B P1 (from the tandem promoter fragment) was 242 ± 31 μM for SC and 361 ± 46 μM for LIN. K_GTP_ for *rrn*B P2 was 62 ± 13 μM for SC and 427 ± 61 μM for LIN (see Appendix A). Similar results were obtained also with *rrn*O P1 and *rrn*O P2 promoters (Appendix A).

Hence, we concluded that for transcription from LIN templates higher concentrations of GTP are needed, regardless of the promoter. The increased K_GTP_ of P*veg* suggested that this change in RNAP affinity for the substrate iNTP might be responsible, at least in part, for the decrease in its activity during the transition from exponential to stationary phase. However, the moderate increase in K_GTP_ of *rrn*B P1 suggested that other factor(s) must be involved in the decrease of this promoter’s activity in vivo. A likely candidate factor was the affinity of RNAP for promoter DNA, i.e., formation of the closed complex or/and the intracellular level of RNAP.

### 3.5. Pveg and rRNA Promoter Affinities for RNAP Change with DNA Relaxation In Vitro

We tested the relative affinity of RNAP for promoter DNA by performing in vitro transcriptions as a function of increasing promoter DNA concentration. We used the tandem *rrn*B P1+P2 DNA fragment and P*veg*. The GTP concentration was set to 1 mM to ensure high efficiency of open complex formation for the tested promoters. Affinity for RNAP of both rRNA promoters was unchanged or slightly decreased on relaxed templates, but this effect was not statistically significant (Figure 4). Therefore, it is possible that the observed decrease in bulk transcription from *rrn*B P1 (SC vs LIN) in vitro could be due to yet another factor (e.g., promoter escape). 

The opposite trend was observed with P*veg*: a relatively low level of the relaxed promoter DNA was able to saturate RNAP compared to the supercoiled template. This behavior could then explain why the activity of P*veg* decreased less than the activity of *rrn*B P1 both in vitro and in vivo. Importantly, it was previously reported that the levels of RNAP subunits decrease from exponential to stationary phase [50,51] and we also observed this trend (Figure 5).

### 3.6. The Effect of Supercoiling on Transcription In Vitro with Alternative Sigma Factors

To extend the study, we tested the effect of supercoiling on transcription from promoters dependent on alternative sigma factors: σ^B^, σ^D^, σ^E^, σ^F^ and σ^H^. σ^B^ is a general stress response sigma factor [52,53], σ^D^ transcribes genes linked with the cell motility and flagella formation [54]. σ^E^ and σ^F^ are sigma factors of early sporulation [55,56]. σ^H^ is responsible for transcription of early stationary genes [57]. 

We tested also σ^N^ (ZpdN) that is present only in the *B. subtilis* NCIB 3610 strain. This strain possesses a large, low-copy-number plasmid pBS32, which was lost during domestication of the commonly used laboratory strains [58]. pBS32 carries genes responsible for cell death after mitomycin C (MMC) treatment, and this effect is dependent on σ^N^ [24,25]. MMC is an antitumor antibiotic that induces DNA strand scission by DNA alkylation leading to crosslinking [59,60,61]. This DNA damage could lead to the formation of linear DNA fragments.

Sequences of respective promoters are listed in Appendix A. We performed transcriptions in vitro on SC and LIN DNA templates with saturating concentration of iNTP. In all but one cases it was the SC DNA that was the better template for transcription, similarly to what we observed with σ^A^ (Figure 6). 

The exception was σ^N^, which displayed about the same or higher activity on LIN DNA than on SC DNA, depending on the promoter (Figure 6B). To show that this effect was not due to some unknown properties of the plasmid DNA bearing these promoters, we also tested a longer *sigN* promoter construct (*sigN* P2+P3). This construct contains σ^A^-dependent *sigN* P2 and σ^Ν^-dependent *sigN* P3 promoters [25] and allowed us to test the effect of SC vs LIN topology for two sigmas with the same template. The results are shown in Figure 6C: σ^A^-dependent P2 is more active on SC DNA whereas σ^N^-dependent P3 prefers LIN DNA for efficient transcription.

## 4. Discussion

In this study we have identified the supercoiling level of DNA as a factor affecting the ability of *Bacillus subtilis* RNAP to transcribe from σ^A^-dependent rRNA promoters as well as from selected promoters depending on alternative σ factors.

### 4.1. rRNA Promoters and Pveg

In our experiments, the drop in rRNA promoter activity during transition to stationary phase was pronounced and concurrent with the onset of stationary phase. A decrease in the activity of *B. subtilis* rRNA promoters in stationary phase was also observed in [62]. However, they used promoter constructs fused with GFP and monitored promoter activity by measuring the intensity of fluorescent signal. GFP is relatively stable, so the decreases they reported were less pronounced than those observed in our experiments.

Here, we propose an updated model of regulation of *B. subtilis* rRNA promoters, revealing supercoiling as a factor involved in their control. The more negatively supercoiled DNA in exponential phase contributes to the high activity of *B. subtilis* rRNA promoters. As this negative supercoiling becomes more relaxed when the cell transitions into stationary phase, this likely contributes to the decrease in the activity of RNAP at rRNA promoters. This is in part due to the decreased affinity of RNAP at rRNA promoters for the initiating GTP but also to a so far unknown step during transcription initiation (e.g., isomerization, promoter escape). The activity of rRNA promoters in stationary phase is also likely affected by the decreased RNAP concentration. The decrease in the available RNAP pool is further exacerbated by the association of the RNAP:σ^A^ holoenzyme with 6S-1 RNA that sequesters it in an inactive form in stationary phase [63]. The combined effect results in the shut-off of rRNA synthesis. Previously, for *E. coli* rRNA promoters*,* the decreased stability of the open complex was identified as the main kinetic intermediate affected by supercoiling [64]. We also note that in *S. aureus* in post-exponential growth phase the downregulation of rRNA is independent of ppGpp or NTP pools [17], and it is possible that DNA topology might be a factor contributing to this downregulation.

In *B. subtilis*, correlations between the supercoiling level and rRNA activity could be found also in the forespore. Within the developing spore, DNA becomes more negatively supercoiled then in stationary phase [40] and this correlates with an increase in rRNA activity in the forespore [62]. Interestingly, during novobiocin treatment the GTP level increases in *B. subtilis* and the changes in DNA topology override its stimulatory effect so that the net result is a decrease in the activity of *rrn*B P1. This is the first observation of a situation where the GTP level and rRNA promoter activity do not correlate in *B. subtilis*. We note that supercoiling was also reported to be involved in rRNA expression in yeast although the mechanistic aspects of this regulation are less understood [65].

The activity of the control P*veg* promoter also decreases from exponential to stationary phase but the decrease is not as pronounced as in the case of *rrn*B P1. The decrease in the activity of P*veg* can be attributed, at least in part, to its increased requirement for the concentration of the iNTP when DNA supercoiling relaxes. Nevertheless, the affinity of P*veg* for RNAP seems to increase with DNA relaxation and this likely partially counteracts the negative effect on open complex formation.

### 4.2. Transcription with Selected Alternative σ Factors

Transcription experiments with promoters dependent on alternative σ factors revealed that linear templates are poorer substrates for the majority of them (σ^B^, σ^D^, σ^E^, σ^F^, and σ^H^). This trend was previously reported also for RNAP:σ^H^ transcribing from the *spoIIA* promoter [66]. For forespore-specific σ^F^, this is consistent with the DNA supercoiling increase in the forespore [40]. For σ^H^ and σ^E^ that are active in stationary phase, although activities of respective promoters strongly decreased with reduced supercoiling in vitro, this likely reflects the physiologically relevant requirements for their activities in the cell in stationary phase. Also, the decrease in the level of supercoiling in stationary phase is likely not as extreme as in our in vitro experiments where it was used to better visualize the effects.

The exception was σ^N^, where transcription (SC vs. LIN) is either relatively unaffected or even increased on linear templates. This is likely physiologically important as mitomycin, which induces σ^N^ expression [24], causes also DNA relaxation and σ^N^ may have evolved to be most active under such conditions. The proficiency of RNAP:σ^N^ on linear templates then may stem from the relatively short spacers of σ^N^ dependent promoters (15 bp compared to 17 bp for σ^A^, [67]), analogously to σ^70^ and σ^S^ of *E. coli* where the different σ activities were proposed to be due to preferences for differently DNA supercoiled templates [68,69,70].

## 5. Conclusions

To conclude, our findings extend the current model of rRNA promoter regulation in *B. subtilis* and reveal the effect of supercoiling on transcription with main and alternative σ factors. 

## Figures and Tables

**Figure 1 microorganisms-09-00087-f001:**
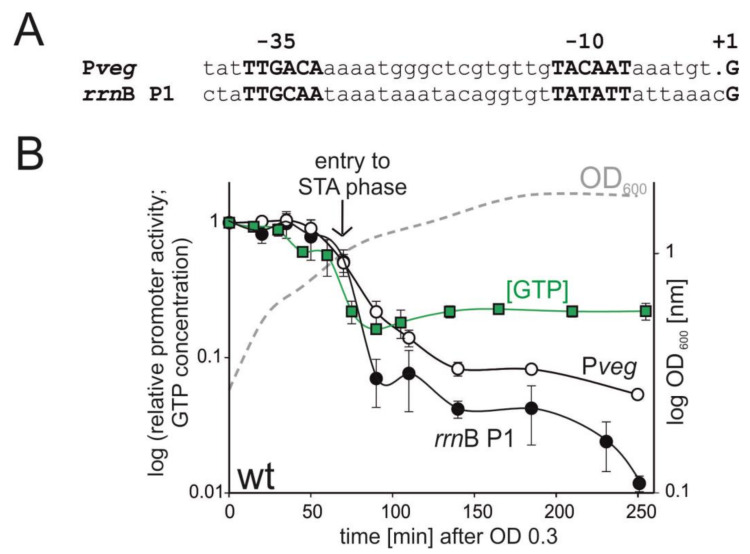
Correlation between GTP concentration and promoter activity after entry into the stationary phase. (**A**) Sequences of P*veg* and *rrn*B P1. (**B**) Relative promoter activities of *rrn*B P1 (black circles) and P*veg* promoters (open circles) after entry into stationary phase, relative GTP concentration (green squares), and optical density (dashed grey line). Promoter activities and GTP concentrations were normalized to 1 at time 0. Promoter activities were measured by qPE from wt *B. subtilis* strains: *rrn*B P1 (LK134), P*veg* (LK135). Promoter activities were calculated from three independent experiments, the error bars show ±SD. The GTP concentrations are from two independent experiments, showing the mean, the bars show the range. A representative bacterial growth curve is shown. The vertical arrow indicates the entry into stationary phase.

**Figure 2 microorganisms-09-00087-f002:**
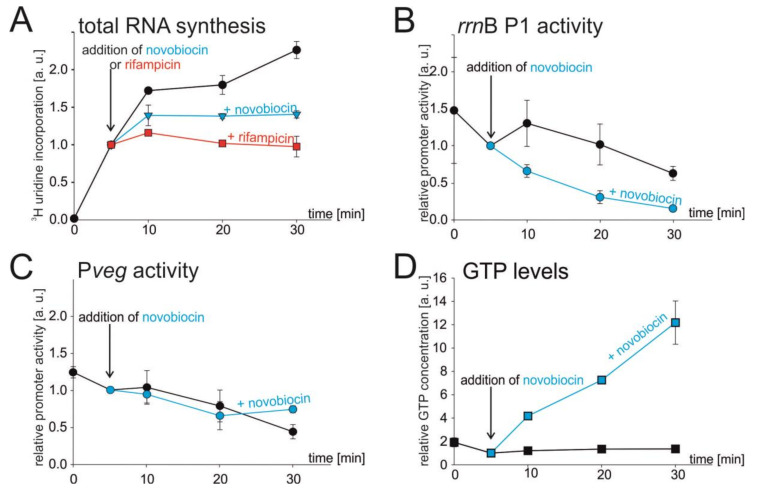
Effect of novobiocin-induced relaxation of chromosome on total RNA synthesis, selected promoter activities, and GTP level. (**A**–**D**) Cells were grown to early exponential phase (OD_600_ ~0.3), and at time 5 min they were treated with novobiocin (5 μg/mL). (**A**) Total RNA synthesis after novobiocin treatment. After ^3^H-uridine had been added (time 0), the culture was divided into three flasks. At time 5 min the cells were treated with novobiocin (blue line) or with rifampicin (red line) as a control, or left untreated (black line). The amount of radiolabeled RNA at 5 min was set as 1. Black circles, mock-treated; blue triangles, treated with novobiocin; red squares, treated with rifampicin. The values are averages of three independent experiments ±SD. (**B**,**C**) The relative activities of *rrn*B P1 and P*veg* promoters after novobiocin treatment. Cells were grown and at 5 min treated with novobiocin or not. RNA was extracted and determination of promoter activity was done by RT-qPCR. Promoter activities were set as 1 at time 5 min. Blue lines are novobiocin-treated samples, black lines are untreated samples. The experiment was performed three times. The error bars show ±SD. (**D**) GTP concentration after novobiocin treatment. Cells were grown in the presence of [^32^P] H_3_PO_4_ and treated with novobiocin. Levels of GTP were determined by TLC. The GTP level at 5 min was set as 1. Results are averages from two measurements. The error bars show the range.

**Figure 3 microorganisms-09-00087-f003:**
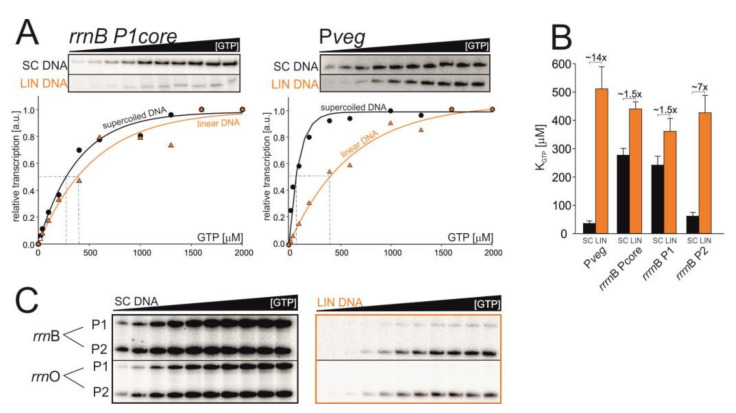
The affinity of RNAP for iNTP in vitro changes on different DNA templates. (**A**) Multiple-round transcriptions as a function of GTP concentration: representative primary data and their graphical comparison for *rrn*B P1core and P*veg*. The maximum signal was set as 1. (**B**) Graphical comparison of K_GTP_ values for SC and LIN DNA templates. The values are calculated from at least four experiments, the error bars show ±SD. (**C**) Low affinity for LIN templates of full-length *rrn* promoter variants. Representative primary data are shown.

**Figure 4 microorganisms-09-00087-f004:**
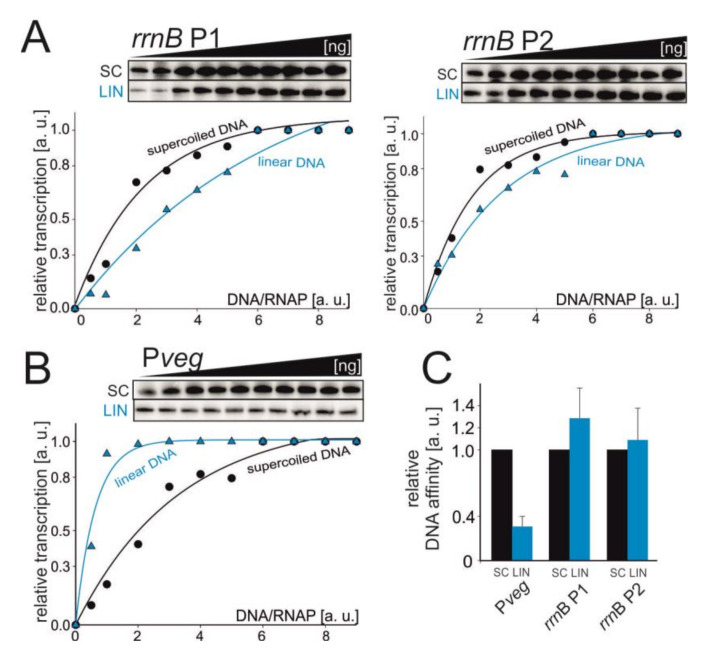
The affinity of RNAP for promoter DNA. Multiple-round transcriptions were carried as a function of the increasing DNA/RNAP ratio. The tested promoters were *rrn*B P1+P2 (**A**) and P*veg* (**B**). Primary data are shown above the graphs. The maximum signal in the plateau phase was set as 1. SC—supercoiled and LIN—linear DNA templates. The experiments were conducted at least four times with similar results. Representative primary data are shown. (**C**) Graphical comparison of relative affinities of RNAP for P*veg* and *rrn*B P1+P2 promoters. The bars show relative concentrations of promoter DNA at which the activity of RNAP was 50%. The affinity of RNAP for SC promoter DNA was set as 1 for each promoter.

**Figure 5 microorganisms-09-00087-f005:**
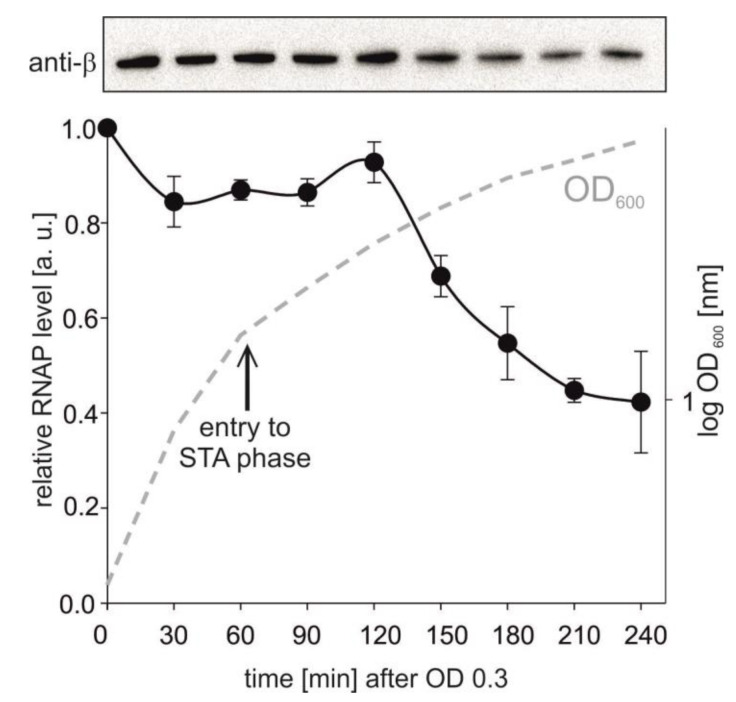
RNAP levels during bacterial growth. Amounts of RNAP were detected by Western blotting from 5 μg of total protein per lane. Representative primary data are shown above the graph. The RNAP level from time point 1 was set as a 1. STA—stationary phase (indicated with the arrow). The experiment was conducted in two independents replicas. The points are averages, the error bars show the range. The dashed line shows a representative bacterial growth.

**Figure 6 microorganisms-09-00087-f006:**
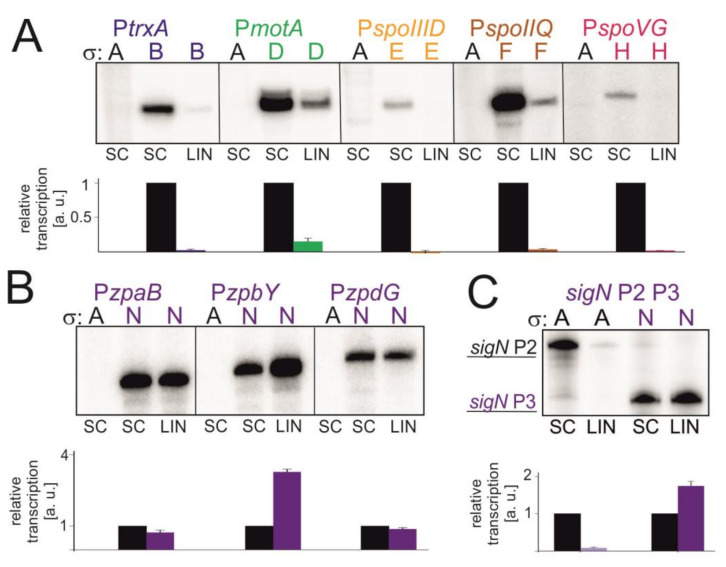
Transcription in vitro with alternative σ factors on different DNA templates. Representative primary data are shown (radioactively labelled transcripts resolved by polyacrylamide electrophoresis). SC stands for supercoiled promoter DNA, LIN for linear DNA. Letters above the gels indicate the sigma factor used—A for σ^A^, B for σ^B^ etc. Each sigma factor is depicted with different color (σ^A^, black; σ^B^, dark blue; σ^D^, green; σ^E^, yellow; σ^F^, brown; σ^H^, red and σ^N^, purple). For each promoter three independent experiments were performed. Transcription from SC was set as 1 for each promoter. Quantitation of results is shown in graphs below the respective primary data. The graphs show averages ±SD. The reactions with σ^A^ on all promoter fragments were used to show that the observed transcription was due to the addition of the specific σ factors and not due to (theoretical) contamination of the core with σ^A^. (**A**) Transcription in vitro from selected σ^B^, σ^D^, σ^E^, σ^F^ and σ^H^ -dependent promoters. (**B**) Transcription in vitro from σ^N^-dependent promoters. (**C**) Transcription in vitro using a longer construct, *sigN* P2+P3. P2 is σ^A^-dependent, P3 is σ^N^-dependent.

**Table 1 microorganisms-09-00087-t001:** Bacterial strains used in a study.

Name	Original code	Construct	Description	Reference
***B. subtilis***				
**LK134**	RLG7554	*rrn*B P1-*lacZ*	MO1099 *amyE*::Cm *rrn*B P1 (−39/+1)-*lacZ*	[11]
**LK135**	RLG7555	P*veg*-*lacZ*	MO1099 *amyE*::Cm P*veg* (−38/−1, +1G)-*lacZ*	[11]
**LK41**	RLG6943	RM-*lacZ*	MO1099 *amyE*::Cm *rrnO* P2 (−77/+50)-*lacZ*	[11]
**LK1723**	RLG7024	*wt* RNAP	β’ with C-ter. His10x; MH5636	[26]
***E. coli***				
**LK22**		SigA	SigA; BL21(DE3)	[27]
**LK1207**		SigB	SigB with C-ter. His6x; BL21(DE3)	This work
**LK1187**		SigD	SigD; BL21(DE3)	[28]
**LK2580**		SigE	SigE with C-ter. His6x; BL21(DE3)	This work
**LK1425**		SigF	SigF with C-ter. His6x; BL21(DE3)	This work
**LK1208**		SigH	SigH with C-ter. His6x; BL21(DE3)	This work
**LK2531**		SigN	His-SUMO-SigN in pBM05; BL21(DE3)	This work
**LK1177**	RLG7558	P*veg*	pRLG770 with P*veg* (−38/+1) +1G; DH5α	[11]
**LK1522**	RLG7596	*rrn*B P1core	pRLG770 with *rrn*B P1 (−39/+1); DH5α	[11]
**LK28**	RLG6927	*rrn*B P1+P2	pRLG770 with *rrn*B P1+P2 (−248/+8); DH5α	[15]
**LK17**	RLG6916	*rrn*O P1+P2	pRLG770 with *rrn*O P1+P2 (−314/+9); DH5α	This work
**LK1231**		P*trxA*	pRLG770 with P*trxA* (−249/+11); DH5α	This work
**LK1233**		P*motA*	pRLG770 with P*motA* (−249/+11); DH5α	This work
**LK2594**		P*spoIIID*	pRLG770 with P*spoIIID* (−150/+10); DH5α	This work
**LK1495**		P*spoIIQ*	pRLG770 with P*spoIIQ* (−251/+9); DH5α	This work
**LK1235**		P*spoVG*	pRLG770 with P*spoVG* (−94/+11); DH5α	This work
**LK2672**		*sigN* P2+P3	pRLG770 with *sigN* P2+P3 (−247/+159); DH5α	This work
**LK2673**		P*zpaB*	pRLG770 with P*zpaB* (−266/+175); DH5α	This work
**LK2608**		P*zpbY*	pRLG770 with P*zpbY* (−304/+155); DH5α	This work
**LK2609**		P*zpdG*	pRLG770 with P*zpdG* (−244/+170); DH5α	This work

**Table 2 microorganisms-09-00087-t002:** List of primers.

Primer No (#)	Sequence 5′→ 3′	
**#1001**	GGAATTCCATATGAATCTACAGAACAACAAGG	Primers for *sigH* cloning into pET-22b(+)
**#1002**	CCGCTCGAGCTATTACAAACTGATTTCGCG	
**#1004**	GGAATTCCATATGACACAACCATCAAAAAC	Primers for *sigB* cloning into pET-22b(+)
**#1006**	CCGCTCGAGCATTAACTCCATCGAGGGATC	
**#1069**	CCGGAATTCATTCCGGAGTCATTCTTACGG	Primers for P*trxA* cloning into pRLG770
**#1070**	CCCAAGCTTCACTGTCATGTACTTTACCATG	
**#1075**	CCGGAATTCCTTTACACTTTTTTAAGGAGG	Primers for P*motA* cloning into pRLG770
**#1076**	CCCAAGCTTCTAGCTTGTCTATGGTTAATATC	
**#1079**	CCGGAATTCTTTATGACCTAATTGTGTAAC	Primers for P*spoVG* cloning into pRLG770
**#1080**	CCCAAGCTTATAAAAGCATTAGTGTATC	
**#1309**	GGAATTCCATATGGATGTGGAGGTTAAGAAAAAC	Primers for *sigF* cloning into pET-22b(+)
**#1311**	CCGCTCGAGGCCATCCGTATGATCCATTTG	
**#1425**	CCGGAATTCCATTCCATCCGGTCTTCAGG	Primers for P*spoIIQ* cloning into pRLG770
**#1426**	CCCAAGCTTCATCACCTCAGCAACATTCTG	
**#2973**	CAGTAACTTCCACAGTAGTTCACCAC	universal reverse primer for PE and qPCR
**#2974**	TCTAAGCTTCTAGGATCCCC	test RNA-specific forward primer for PE and qPCR
**#2975**	GTCGCTTTGAGAGAAGCACA	RM RNA-specific forward primer for PE and qPCR
**#3109**	GCGAATTCCGTGTCGGTCAACATAATAAAGG	Primers for *sigN* P2+P3 cloning into pRLG770
**#3110**	GCAAGCTTCGGCAAAAATCTTTCTCTCACC	
**#3111**	GCGAATTCGCGATGAATGAAGAGACACGG	Primers for P*zpaB* cloning into pRLG770
**#3112**	GCAAGCTTAGTCCATCTCGAAGATCTGGT	
**#3113**	GCGAATTCGACTCCAACATTTCTATTCC	Primers for P*zpbY* cloning into pRLG770
**#3114**	GCAAGCTTGGTCTTCTTCACTTAATTCA	
**#3117**	GCGAATTCTCAAAGATCTTCTAACTTGT	Primers for P*zpdG* cloning into pRLG770
**#3118**	GCAAGCTTGGCAGTAATCAATCAATTCT	
**#3166**	CGGCATATGTACATAGGCGGGAGTGAAGCC	Primers for *sigE* active form cloning into pET-22b(+)
**#3167**	CCGCTCGAGCACCATTTTGTTGAACTCTTTTC	
**#3170**	GGCGAATTCGCTTATTTCATTTTACAGGAG	Primers for P*spoIIID* cloning into pRLG770
**#3171**	CCGAAGCTTTGTTAGGTTTGTAACAGTGT	
**primer A**	GGGAATTCATGGACATCAATGATATCTC	Primers for *rrn*O P1+P2 cloning into pRLG770
**primer B**	GGAAGCTTTCAAAGCGACTACTTAATAG	

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
