# Peer review of "Effects of DNA Topology on Transcription from rRNA Promoters in Bacillus subtilis"

_microorganisms, 2021, doi:10.3390/microorganisms9010087_

Round 1

Reviewer 1 Report

The shut-down of rRNA synthesis in non-growing cell is a conserved mechanism in most/all organism. Previous work of the group has shown that in B. subtilis this is at least in part linked to the pppGpp dependent decrease of the GTP level. The data presented here show that besides GTP regulation other mechanisms must be in place to inhibit rRNA promoter activity in the late growht phase. In a series of experiments they show compelling evidence that promoter activity is correlated to supercoiling and as such might be an important factor to control the coordinated rRNA synthesis during growth. The authors could further show that only one alternative sigma factor prefers a relaxed template. This is an well-written manuscript supported by well designed experiments. I have only some minor comments:   

Line 59: In S. aureus it was already shown that down-regulation of rRNA is only partially correlated to GTP and down-regulation at later growth occurs independent of GTP or (p)ppGpp.

Methods: indicate how pppGpp/ATP was detected. Same as for GTP?

Line 107: Describe how RM was made or give a reference.

Line 269, line 439 : The author make the assumption that supercoiling is growth phase dependent and DNA more relaxed in stationary phase. Although this an attractive hypothesis some firm data to support this notion should be given. In the cited review there is little evidence given and most of the work is based on E. coli. Of note in B. subtilis most of the described supercoiling-interfering factors are not present or different. Thus, this statement should be supported by solid published or own data. Otherwise, the assumption has to be toned down.

Line 454: It would be interesting to see how novobiocin influence the nucleotid pool. Are there similar changes in a relA/spot mutant? This should be discussed.

Author Response

Thank you for your time and review.

Attached is the manuscript with changes in the track changes option. Here, point-by-point responses to your comments follow.

Line 59: In S. aureus it was already shown that down-regulation of rRNA is only partially correlated to GTP and down-regulation at later growth occurs independent of GTP or (p)ppGpp.

RESPONSE

We agree.

ACTION TAKEN

We modified the text accordingly (see page 15, line 461 [all line numbers refer to the “all markup version”]).

Methods: indicate how pppGpp/ATP was detected. Same as for GTP?

RESPONSE

Done.

ACTION TAKEN

We added information about ppGpp and ATP extraction and detection to Methods (section 2.3.)

Line 107: Describe how RM was made or give a reference.

RESPONSE

Done.

ACTION TAKEN

The experiment and RM preparation were done as described in reference 15 (Krásný et al., 2008) – line 106. Nevertheless, to avoid confusion, we added the reference also to line 109, where the preparation of the RM is specified.

Line 269, line 439 : The author make the assumption that supercoiling is growth phase dependent and DNA more relaxed in stationary phase. Although this an attractive hypothesis some firm data to support this notion should be given. In the cited review there is little evidence given and most of the work is based on E. coli. Of note in B. subtilis most of the described supercoiling-interfering factors are not present or different. Thus, this statement should be supported by solid published or own data. Otherwise, the assumption has to be toned down.

RESPONSE

We agree.

ACTION TAKEN

We modified the text and included references for primary publications on E. coli and B. subtilis that support the statement (line 270-271).

Furthermore, we toned down the statement about the effect of supercoiling of rRNA promoter activity (line 450).

Line 454: It would be interesting to see how novobiocin influence the nucleotid pool. Are there similar changes in a relA/spot mutant? This should be discussed.

RESPONSE

We agree. The increase in the GTP level itself was already a surprise. We have not performed experiments with a pppGpp minus strain but it is a good point and of interest for future studies.

ACTION TAKEN

We added a sentence at the end of the novobiocin section (3.3.; line 317) and a sentence at the beginning of section 3.4. (see also comment 1 of reviewer #2).

Reviewer 2 Report

The submission by the Krásny lab is an interesting contribution to our understanding of rRNA transcriptional regulation in a Gram-positive model organism. The paper is well documented. The experiments are well designed, the appropriate controls are included but I found several points to be unclear. I suspect that most of this could be fixed by changes in the text.

  1. The in vivo experiments with novobiocin are suggestive. However, adding an inhibitor with a major effect to growing cells is fraught and the authors should perhaps point out that the effects shown in Fig. 2 might be due to an indirect mechanism rather than a direct effect of supercoiling on the various promoters, or even to an off-target effect of the antibiotic. A resistant gyrase mutant might be a nice control, but not essential. I think qualifying the conclusions of these experiments would improve the flow of the manuscript and set the reader up for the in vitro experiments.
  2. In the legend to Fig. 2 it is stated that “Promoter activities were set to 1 at time 0”. But in panels B and C the initial values are higher than 1.
  3. I do not see an explanation of the different effects of relaxation on rRNA vs the veg promoters when yield of transcript is measured. The affinity for veg is affected strongly but the yield is relatively unaffected. The affinity of P1 for iGTP is slightly affected by relaxation but the maximum yield of transcript is dramatically decreased. Later (lines 363-364) it is suggested that a decreased affinity of relaxed rRNA promoters for RNAP may be the cause of this large effect. But this is essentially refuted in the next section. There is something unclear here, and perhaps it is in the writing.
  4. It is stated that “Affinity for RNAP of both rRNA promoters was unchanged or slightly decreased on relaxed templates”. Indeed, the effect is quite small, not more than ~20% with appreciable error bars. I am not sure that this is a significant effect or if it is biologically important, in agreement with the stated possibility that affinity might be “unchanged”. Can the authors test the significance of this effect statistically?
  5. Line 400 and following. There is no reference here, which prompted me to look for it. The Kearns lab paper should be cited.
  6. SigH, SigE and SigF are all needed for spore formation. If stationary phase is accompanied by relaxation, why are these promoters down-regulated in vitro by linearization? I do not doubt the data, but some discussion or comment might be interesting.

Author Response

Thank you for your time and review.

Attached is the manuscript with changes in the track changes option. Here, point-by-point responses to your comments follow.

  1. The in vivo experiments with novobiocin are suggestive. However, adding an inhibitor with a major effect to growing cells is fraught and the authors should perhaps point out that the effects shown in Fig. 2 might be due to an indirect mechanism rather than a direct effect of supercoiling on the various promoters, or even to an off-target effect of the antibiotic. A resistant gyrase mutant might be a nice control, but not essential. I think qualifying the conclusions of these experiments would improve the flow of the manuscript and set the reader up for the in vitro experiments.

RESPONSE

Thanks, this is a good point!

ACTION TAKEN

We have changed the text accordingly - we added a sentence at the end of the novobiocin section (3.3.; line 317 [all line numbers refer to the “all markup version”]) and a sentence at the beginning of section 3.4. (see also the last comment of reviewer #1).

2. In the legend to Fig. 2 it is stated that “Promoter activities were set to 1 at time 0”. But in panels B and C the initial values are higher than 1.

RESPONSE

Thank you for the pointing out. The correct statement is “Promoter activities were set to 1 at time 5 min”.

ACTION TAKEN

We corrected this issue in the text (line 301).

3. I do not see an explanation of the different effects of relaxation on rRNA vs the veg promoters when yield of transcript is measured. The affinity for veg is affected strongly but the yield is relatively unaffected. The affinity of P1 for iGTP is slightly affected by relaxation but the maximum yield of transcript is dramatically decreased. Later (lines 363-364) it is suggested that a decreased affinity of relaxed rRNA promoters for RNAP may be the cause of this large effect. But this is essentially refuted in the next section. There is something unclear here, and perhaps it is in the writing.

RESPONSE

We agree. It is not clear, what is responsible (which step during the initiation process) for the decrease in bulk transcription of rrnB P1 (SC vs LIN) in vitro. We speculate that it might be an effect at another stage of transcription (e. g. promoter escape). Nevertheless, as the affinity of rrnB P1 for RNAP does not change much (if at all) with changes in DNA topology, and the RNAP level decreases in the cell in stationary phase, this decrease in the RNAP level then likely contributes to the decrease in the rrnB P1 activity in vivo.

ACTION TAKEN

We added sentences addressing this issue to line 377 in the Results section and to line 452 in Discussion. We clarified that besides the affinity for the initiation GTP, another step(s) during transcription initiation are likely also involved.

4. It is stated that “Affinity for RNAP of both rRNA promoters was unchanged or slightly decreased on relaxed templates”. Indeed, the effect is quite small, not more than ~20% with appreciable error bars. I am not sure that this is a significant effect or if it is biologically important, in agreement with the stated possibility that affinity might be “unchanged”. Can the authors test the significance of this effect statistically?

RESPONSE

We agree.

ACTION TAKEN

The difference is not statistically significant. We changed the phrasing, line 377.

5. Line 400 and following. There is no reference here, which prompted me to look for it. The Kearns lab pa`per should be cited.

RESPONSE

The references were on line 403 – Ref. 24, 25 and 57 (Maygmarjav et at., 2016; Burton et al., 2019 and Earl et al., 2007).

ACTION TAKEN

To make it more clear, we left reference 57 (now No. 58) after the sentence ending with “….commonly used laboratory strains.” (line 410) and moved references to Kearns lab papers (Ref Nos. 24, 25) after the sentence ending with “…this effect is dependent on sN” (line 411).

6. SigH, SigE and SigF are all needed for spore formation. If stationary phase is accompanied by relaxation, why are these promoters down-regulated in vitro by linearization? I do not doubt the data, but some discussion or comment might be interesting.

RESPONSE

We agree. The activities of these promoters are adapted to the relevant physiological situation (stationary phase). For SigF, the results are consistent as the level of supercoiling increases in the forespore. For SigH and SigE that are active in stationary phase – although activities of respective promoters strongly decreased with reduced supercoiling in vitro – this likely reflects the physiologically relevant requirements for their activities in the cell in stationary phase. Also, the decrease in the level of supercoiling in stationary phase is likely not as extreme as in our in vitro experiments where it was used to better visualize the effects.

ACTION TAKEN

We changed the text accordingly – starting on line 480.
